# Ferroptosis in Cancer: Mechanism and Therapeutic Potential

**DOI:** 10.3390/ijms26083852

**Published:** 2025-04-18

**Authors:** Mansaa Singh, Hasmiq L. Arora, Rutuja Naik, Shravani Joshi, Kaveri Sonawane, Nilesh Kumar Sharma, Birandra K. Sinha

**Affiliations:** 1Cancer and Translational Research Lab, Dr. D.Y. Patil Biotechnology & Bioinformatics Institute, Dr. D.Y. Patil Vidyapeeth, Pune 411033, India; mansaasingh.1258@gmail.com (M.S.); hasmiqliba21@gmail.com (H.L.A.); rutujanaik08@gmail.com (R.N.); shravanijoshi11@gmail.com (S.J.); kaverisonawane004@gmail.com (K.S.); 2Mechanistic Toxicology Branch, Division of Translational Toxicology, National Institute of Environmental Health Sciences, Research Triangle Park, Durham, NC 27709, USA

**Keywords:** ferroptosis, cell death, signaling, oxidative stress, iron, lipid peroxidation

## Abstract

Cancer drug resistance occurs when cancer cells evade cell death following treatment with chemotherapy, radiation therapy, and targeted therapies. This resistance is often linked to the reprogramming of programmed cell death (PCD) pathways, allowing cancer cells to survive drug-induced stress. However, certain anticancer therapies, when combined with specific agents or inhibitors, can induce ferroptosis—a form of cell death driven by iron-dependent lipid peroxidation. Currently, extensive preclinical and clinical research is underway to investigate the molecular, cellular, and tissue-specific mechanisms underlying ferroptosis, with the goal of identifying strategies to overcome drug resistance in cancers unresponsive to conventional PCD pathways. By harnessing ferroptosis, cancer cells can be compelled to undergo lipid peroxidation-induced death, potentially improving therapeutic outcomes in patients with cancer. This short review aims to enhance the understanding of ferroptosis inducers in cancer therapy and stimulate further research into ferroptosis-based approaches for more effective clinical cancer treatment.

## 1. Introduction

Tumor heterogeneity (TMH) refers to the diversity of tumor cells within a tumor or among different tumors. Even within a particular tumor, the cells differ in morphology, proliferation, metastatic potential, gene expression patterns, and metabolic requirements. TMH exists at the molecular, cellular, and tissue levels and is responsible for survival, cell death, and resistance to cancer drugs [1,2]. Resistance to conventional anticancer drugs is attributed to the complex and evolving nature of tumors [3,4]. The emergence of drug resistance in cancer cells provides distinctive advantages, such as proliferation, survival, evasion of immune destruction, and plasticity [3,4].

Ferroptosis is a distinct form of regulated cell death that involves cellular lipid peroxidation and iron metabolism [3,4,5]. Lipid metabolic pathways that generate lipid peroxides, when combined with excess iron ions, trigger ferroptosis type of cell death. Lipid peroxides accumulate when their removal mechanisms, such as the glutathione-dependent antioxidant system (GPX4 enzyme), are impaired. This has potential therapeutic implications for cancer treatment [6,7]. Ferroptosis is a form of caspase-independent programmed cell death driven by the mitochondrial and membrane-associated accumulation of reactive oxygen species and lipid peroxides [8,9,10,11]. Understanding the molecular mechanisms of ferroptosis, including iron metabolism, lipid peroxidation, and antioxidant defenses, could help develop combination therapies with existing cancer treatments [12,13,14,15,16,17].

In this review, we discuss ferroptosis from the perspective of its use as a novel strategy for combinatorial anticancer drug development at both preclinical and clinical levels. We hope that this discussion will stimulate further research into ferroptosis-based approaches for more effective cancer treatment in clinical practice.

## 2. Types of Drug-Induced Cell Death

The acquisition of functional capacities by human cells during the transition from normal to neoplastic development stages is attributed to many critical characteristics known as tumor hallmarks [18,19]. Among the key hallmarks of tumor, the interdependence and crosstalk between cancer cell drug resistance and metabolic reprogramming have been extensively studied and determined to provide several advantages to cancer cells in a drug-induced environment [20,21,22,23]. Therefore, therapeutic management of deregulated cellular metabolism, such as lipid metabolism [24,25], oxidative stress [26], iron-induced ROS formation [27,28], and overcoming resistance, is being explored at preclinical and clinical levels [29,30].

### 2.1. Apoptosis

Cancer therapies target apoptosis, a programmed form of cell death [31]. However, mutations in apoptotic pathways promote treatment resistance [32,33], allowing cancer cells to survive and proliferate [34,35]. Alternative cell death mechanisms like necrosis [36], senescence [37], autophagy [38], and mitotic catastrophe [39] enable further evasion, presenting potential drug targets [40,41,42,43]. Mutations in caspases [44], p53 [45], and Bcl-2 proteins [46] weaken apoptotic control, driving unregulated growth. Cancer cells enhance their survival by upregulating anti-apoptotic proteins, mutating pro-apoptotic genes, and altering pathways like PI3K/Akt and NF-κB [47,48]. Apoptosis maintains tissue homeostasis by removing damaged cells [49]; however, frequent mutations in p53 [50,51] and overexpression of Bcl-2 proteins [52] inhibit this process. Caspases, which are essential for apoptosis, are often suppressed in cancer cells [53,54]. Traditional therapies induce apoptosis through intrinsic and extrinsic pathways [55], involving caspase activation [56] and cellular changes like chromatin condensation [57]. While apoptosis relies on a balance between pro- and anti-apoptotic factors [58], mutations in TP53 [59] often disrupt this process. Many chemotherapeutic and targeted drugs function by inducing apoptosis [60].

### 2.2. Autophagy

Autophagy is a process in which cells digest and recycle their constituents [61]. Autophagy generally promotes cell survival by supplying nutrients during stress and maintaining homeostasis [62]. However, autophagy can also cause cell death, especially in apoptosis-resistant cancer cells [63]. Certain cancer therapies modulate autophagy to facilitate death in cancer cells [64]. For instance, a few mTOR inhibitors have been shown to induce autophagy, resulting in tumor cell death, especially when used in combination with other modalities of treatment [65]. Autophagy can act as a survival mechanism and as a form of cell death [66]. Autophagy has been reported to be regulated by many signaling pathways, including PI3K/Akt/mTOR [67], and often results from responses to metabolic and therapeutic stressors [68]. Pharmacological agents are known to modulate autophagy and cause cell death in apoptosis-resistant cancer cells [69].

### 2.3. Necrosis

Necrosis is uncontrolled cell death or a form of accidental cell death caused by extreme stress or damage, where the cell membrane bursts, leading to inflammation [70]. Although necrosis is generally not targeted in cancer therapy, some treatments may cause unintended necrosis within tumors, resulting in self-destruction [71]. The process of necrosis normally occurs due to overwhelming cellular stress that negatively affects mitochondrial function, resulting in energy depletion and activation of proteases, leading to cellular membrane rupture [72]. Programmed necrosis may be induced via mediators involving RIP kinases [73], PARP [74], and TLR-4 signaling pathways [75], particularly those mediated in response to stimuli, including graphene-based materials [76]. The induction of both apoptosis and autophagy can prevent necrosis, demonstrating that cell death pathways are delicately balanced, where a major part does not prevent but induces counter-cell-death pathways [77].

### 2.4. Necroptosis

Necroptosis, or regulated necrosis, is a type of cell death characterized by the disintegration of the plasma membrane and oncosis (swelling of subcellular organelles) without displaying the unique features of apoptosis or autophagy [78]. It is induced by the kinase activity of receptor-interacting serine/threonine kinase protein (RIPK) 1, thereby forming complex IIB and causing cell necroptosis; RIPK 1 and RIPK 3 are responsible for the phosphorylation of mixed-lineage kinase-like (MLKL) [79,80]. The activation of cell surface death receptors (such as FasRs, TNFR1, IFN receptors, and TLRs) and RNA- and DNA-detecting molecules in cells initiates the necroptotic process [78,79,80]. Studies on drug-induced cell death have revealed the complex mechanisms through which apoptosis, autophagy, and necrosis are regulated and connected [81]. In addition to these known cell death pathways, ferroptosis offers a way to bypass these traditional forms of cell death, utilizing iron-catalyzed lipid peroxidation as its killing mechanism. These are vital for understanding cancer therapy and the development of new therapeutic approaches.

### 2.5. Ferroptosis

Ferroptosis is an iron-dependent, non-apoptotic form of regulated cell death driven by lipid peroxidation [82]. First identified by Dixon et al. [6], it differs from apoptosis, necrosis, and autophagy in terms of structure and function and is gaining therapeutic interest [80,81,82,83,84,85]. Cell fate is influenced by oxidative stress, which is a key factor in metabolism and survival [86,87]. Various environmental and genetic factors, including heat [88], radiation [89], metabolism [90], redox homeostasis [91], immune surveillance [92], and oncogenic signaling [93] contribute to oxidative stress, triggering ferroptosis through unchecked lipid peroxidation [94]. Ferroptosis is characterized by mitochondrial shrinkage, outer membrane damage, increased membrane density, and reduced NADH levels, without chromatin condensation [95,96,97,98,99,100]. Unlike apoptosis, it does not require caspase activation [101] and is unaffected by necrosis regulators RIP1/RIP3 and Cyclophilin D [102] or autophagy inhibition by 3-MA [103], confirming its distinct nature [104]. Identified as a potential cancer therapy, ferroptosis can target resistant cells through iron-catalyzed lipid peroxidation [104,105,106]. It results from iron overload [107] and metabolic imbalance [108] and is regulated by genes such as GPX4 [109] and ACSL4 [110], which control lipid metabolism and antioxidant defense [111].

### 2.6. Ferroptosis: Molecular Mechanism

Ferroptosis is an iron-dependent, intrinsically regulated cell death that is distinct from apoptosis and necrosis based on iron dependence and the accumulation of lipid peroxides [112]. These processes are complex and interwoven with iron metabolism, lipid peroxidation, and antioxidant defense mechanisms [113]. Ferroptosis requires labile iron, which catalyzes the formation of lipid hydroperoxides, mainly through the action of lipoxygenases [114]. The antioxidant enzyme GPX4 utilizes GSH to regulate ferroptosis by reducing lipid hydroperoxides [115]. When homeostasis is disturbed or antioxidant defenses are compromised, lipid hydroperoxides accumulate, causing cell death [116]. Therefore, it is important to understand the pathways involved in ferroptosis, as therapeutics may be applied to diseases associated with ferroptosis [117].

Ferroptosis, a form of regulated cell death, is characterized by the formation of ●OH generated from the reaction of H_2_O_2_ with Fe^2+^ (the Fenton reaction) [118]. Ferroptosis is triggered by lipid peroxidation, which is driven by hydroxyl radicals (●OH) reacting with polyunsaturated fatty acids (PUFAs) in the cell membrane. This process generates lipid hydroperoxides, which accumulate if not neutralized by antioxidant systems, such as glutathione peroxidase 4 (GPX4), leading to membrane rupture and cell death [119].

Iron can catalyze ferroptosis both enzymatically and non-enzymatically. Enzymatic catalysis involves facilitating the oxidation of lipids within cells, forming lipid peroxides [120]. Autophagic degradation of ferritin (an iron-storage protein) results in the leakage of free iron (Fe^2+^) into the cytoplasm, thereby expanding the pool of redox-active iron and enhancing susceptibility to ferroptosis. The overexpression of transferrin receptor 1 (TfR1) enhances the amount of iron that is internalized and enhances susceptibility to ferroptosis [121]. The main factors involved in the process of lipid peroxidation include lipoxygenases and phosphorylase kinase G2 (PHKG2), which regulate the availability of iron required to catalyze the peroxidation of polyunsaturated fatty acids (PUFAs) [122].

In addition to GPX4, the cystine-glutamate antiporter SLC7A11 and ferroptosis suppressor protein 1 (FSP1) contribute significantly to the inhibition of ferroptosis via lipid peroxide detoxification [123]. Both genetic alterations and chemical inhibition of antioxidant defenses trigger the induction of ferroptosis [124]. The main sources of ROS generation during ferroptosis are activated mitochondria, NADPH oxidases, and the Fenton reaction [125]. Lipid peroxidation increases the levels of lipid ROS, forming a vicious cycle that increases cell death [126]. Mitochondria are a major source of ROS, and disruption of iron metabolism can trigger ferroptosis [125]. Inhibition of mitochondrial oxidative phosphorylation further increases intracellular ROS levels [120,127,128,129]. A flow model of the steps in ferroptosis involving various membrane and intracellular components including transferrin receptors, glutamine-cystine antiporter (Xc), LOX, GPX, GSH, iron, and lipid peroxides, is presented (Figure 1). A summary of the various molecular mechanisms of ferroptosis is presented, including details such as genes and pathways (Table 1 and Figure 2).

## 3. Types of Ferroptosis

Ferroptosis is traditionally considered a uniform cell death process involving multiple regulatory pathways [127,128]. However, distinct mechanistic variations exist, reflecting how cells manage oxidative stress, iron overload, or lipid peroxidation, suggesting pathway-dependent or context-specific subtypes [129,130].

Classical ferroptosis is the most thoroughly studied form, in which ferroptosis is initiated through lipid peroxidation and iron-catalyzed reactions and failure of the antioxidant defense system, primarily glutathione peroxidase 4 (GPX4). The major contributors are iron overload and glutathione depletion, with the accumulation of lipid peroxides [124,125]. Under conditions of GSH depletion, such as during oxidative stress or inhibition of the cystine-glutamate antiporter (system Xc−), GPX4 is also inhibited. This results in the accumulation of lipid peroxides and eventual ferroptotic cell death [129]. Iron plays a central role in catalyzing Fenton reactions to produce hydroxyl radicals that intensify lipid peroxidation [131]. Whereas glutathione plays a critical role as a cofactor for GPX4, an antioxidant that neutralizes ROS and lipid ROS [132].

Mitochondria-dependent ferroptosis is implicated in the promotion of oxidative damage due to mitochondria dysfunction [133]. Mitochondria are the power source for cells, which are not only essential for energy production but also for regulating respiration and the production of mitochondria-specific ROS, which accordingly heightens susceptibility to ferroptosis [134]. This pathway can be triggered by the disruption of mitochondria function due to various stressors, iron accumulation, or specific inhibitors of gamma-glutamylcysteine synthase, such as buthionine sulfoximine (BSO), which depletes GSH, destroys mitochondrial functions, and elevates ROS levels. This ultimately leads to an increase in ROS and oxidative damage to cellular lipids, inducing ferroptosis through a mitochondria-dependent pathway.

FSP1, also known as ferroptosis suppressor protein1, has been described as a mechanism of protecting cells from ferroptosis, independent of the activities of GPX4 [135]. It contains CoQ10 (ubiquinone), a lipid-soluble antioxidant that prevents lipid peroxidation, recycles CoQ10, thereby regenerating its antioxidant capacity and providing an additional defense mechanism. Cells lacking functioning GPX4 may undergo ferroptosis through the FSP1 pathway [136].

NADPH oxidase (NOX)-dependent ferroptosis is associated with NOX catalytic activity, leading to the generation of ROS by transferring electrons from NADPH to oxygen, producing superoxide and ROS, which damage cellular membranes [137]. NADPH oxidase enzymes produce ROS, leading to lipid peroxidation and ferroptosis [138]. There is a link between NOX activity and the induction of ferroptosis, especially via the mediation of the YAP-TAZ signaling pathway [139]. NOX inhibitors are suggested to inhibit ferroptosis, especially in cells with high NOX activity. The enhancement of NOX activity can lead to an increase in ferroptosis [140]. NOX-dependent ferroptosis may function as a distinct regulatory mechanism, offering potential therapeutic applications in diseases with disrupted NOX activity.

Iron overload-induced ferroptosis results from unregulated oxidative damage via the Fenton reaction, in which iron catalyzes hydroxyl radical formation, triggering lipid peroxidation. This occurs in cells lacking proper iron storage mechanisms, such as ferritin levels [141]. This kind of ferroptosis is due to imbalanced iron homeostasis, for example, through ferritin degradation, induction of the transferrin receptor, or accumulation of excessive iron [142]. Chelators of iron, such as deferoxamine, also repress it by reducing free iron concentrations and preventing the Fenton reaction from occurring [143].

Erastin-induced ferroptosis is linked to the blockade of system Xc−, an antiporter of cystine/glutamate that plays a major role in maintaining cellular glutathione levels [144]. Erastin disrupts the synthesis of glutathione by inhibiting cysteine uptake, leading to the depletion of this antioxidant [145]. Glutathione serves as a cofactor for GPX4 and is required for its activity of GPX4 [146]. This pathway is important because it links the disruption of cellular antioxidant defenses with ferroptosis initiation. It has major applications in cancer treatment because it can enhance the efficacy of chemotherapy and other treatments.

Radiation-induced ferroptosis has recently been indicated to cause ferroptosis via exposure to ionizing radiation, leading to the generation of ROS, particularly lipid peroxidation of iron-loaded cells [147]. This mechanism remains of significant interest in cancer therapy, given that the induction of ferroptosis renders radiation therapy more potent [148]. Irradiation prompts iron-dependent generation of ROS and subsequent lipid peroxidation [149]. A summary of the various types of ferroptosis is provided, including descriptions of key genes and cancer hallmarks (Table 2 and Figure 3).

### 3.1. Ferroptosis: Detection, Estimation, and Biomarkers of Ferroptosis

The detection and estimation of ferroptosis assume major importance in understanding its role in such diseases and devising therapeutic strategies [150,151,152,153,154,155,156]. A particular fluorescent probe, PPAC-C4, has been developed for the dual ratio and ultrahigh-accuracy quantification of mitochondrial viscosity, which is enhanced during ferroptosis [150]. To track the reversal of polarity in the plasma membrane during ferroptosis, polarized fluorescent probes Mem-C1C18 and Mem-C18C18 were developed. In high-resolution fluorescence labeling and quantification, Mem-C1C18 was shown to be superior to C18C18 [151].

Detecting ferroptosis in cells requires specific biomarkers that are unique to lipid peroxidation and iron metabolism [152]. Common methods of detection include measuring lipid peroxidation products, such as malondialdehyde (MDA) and 4-hydroxynonenal (4-HNE), and using iron-sensitive dyes to detect intracellular iron [153]. Measuring these markers can help detect cells in the process of ferroptosis [154]. The cystine/glutamate antiporter SLC7A11 (also commonly known as xCT) is reported to mediate metabolic reprogramming in cancer by impinging on ferroptosis-dependent cell death pathways [155].

In recent years, the identification of biomarkers of ferroptosis has attracted great attention, as they can significantly enhance chemotherapy by providing tools to monitor, predict, and potentially mitigate cell death related to iron-dependent lipid peroxidation. Several oxidative stress biomarkers have now been identified that appear to be present during ferroptosis. For example, ophthalamic acid has been detected in blood following depletion of GSH in the liver and is now considered a biomarker of oxidative stress [156]. We have also detected the presence of opthalamates during the process of ferroptosis in ovarian cancer cells [157]. In addition, modulation of carnitine levels has been observed during ferroptosis in both colon and ovarian cells [158]. Various genes are induced during oxidative stress that are also induced during ferroptosis. Of particular interest is the CHAC1 gene, a master regulator of oxidative stress [159], which is significantly induced during ferroptosis in colon [158] and ovarian cancer cells [160].

Advancements in fluorescent probes and multiplexed assays have significantly improved ferroptosis detection [161,162]. Precise monitoring of molecular targets enhances the understanding of their role in disease and aids in the development of targeted therapy [163]. Organic fluorescent probes can detect multiple biomolecules and microenvironments during ferroptosis, offering non-destructive and easy-to-prepare tools for assessing homeostasis and physiological changes [162,163].

### 3.2. Ferroptosis: Preclinical Evidence

Preclinical and clinical researchers have generated significant interest in modulating ferroptosis in diseases such as cancer as a new avenue of therapy [81,82,83,84,85,86,87,88,89,90]. Combining chemotherapy with ferroptosis inducers is believed to be an improved strategy to overcome chemotherapy resistance and enhance therapeutic efficacy [164,165,166]. It is now recognized as a promising avenue for cancer treatment in the clinic [167,168,169,170,171,172,173,174,175,176]. PRLX93936, an analog of erastin, has been tested in clinical trials. Co-treatment with cisplatin and PRLX93936 induces lipid peroxidation and Fe^2+^ production, thereby promoting ferroptosis [165].

Ferroptosis inducers, such as erastin and its analogs, have shown promise in sensitizing resistant tumor cells to various chemotherapeutic agents [166,167,168,169,170,171,172]. Our recent work has shown that erastin, as a single agent, is highly effective against both P-gp- and BCRP-expressing cell lines [164,167]. However, RSL3, an inhibitor of GPX4, was significantly more cytotoxic to these cells [167]. Furthermore, both erastin and RSL3 significantly enhanced the cytotoxicity of adriamycin and topotecan in these cell lines [167]. Yang et al. reported that CHAC1 (glutathione-specific γ-glutamylcyclotransferase 1) induction significantly enhanced radiation-dependent ferroptosis in thyroid cancer cells [168]. Recently, ML162 and ML210 have been indicated to lack an inhibitory effect on selenoprotein GPX4. However, ML162 and ML210 were found to be efficient inhibitors of another selenoprotein, TXNRD1 [169].

Inducers of ferroptosis are also implicated in the modulation of tumor responses, including immune responses [170]. Mitigation of GPX4 by lentivirus sh-GPX4 in TNBC cells enhances sensitivity to gefitinib by modulating ferroptosis [171]. In cisplatin-resistant ovarian cancer cells, tripterygium glycosides disturb redox homeostasis and enhance ferroptosis, leading to increased chemosensitivity [172]. In ovarian cancer cells, the combinatorial effects of NRF2 inhibitors and GPX4 inhibitors resulted in the suppression of growth and decreased formation of spheroids over anticancer drugs alone in cell culture and 3D model, respectively [173]. In HCC cells, silencing of soluble vector family member 6 (SLC2A6) leads to the suppression of proliferation, migration, and invasion, and these observations are linked to the ferroptosis pathway [174]. Additionally, the combination of asiatic acid and sorafenib enhances ferroptosis in HCC cells by mediating JNK1/2 signaling [175].

A summary of data on the various inducers of ferroptosis in combination with anticancer drugs is presented (Table 3).

### 3.3. Ferroptosis: Clinical Evidence

Ferroptosis has emerged as a promising strategy for cancer treatment in clinical settings [167,168,169,170,171,172,173]. However, most clinical applications remain in the investigation stage. Among the most encouraging directions are preclinical studies showing synergistic effects between ferroptosis inducers and existing chemotherapeutic agents, highlighting the potential of combination therapy [165,166]. While robust preclinical data support ferroptosis as a viable anticancer approach, its translation into clinical practice is still in the early stages, albeit gaining momentum. Clinical evidence of the induction of ferroptosis in human diseases such as epilepsy, mucositis, and autoimmune diseases has been recorded in the Clinical Trial Registry. However, clinical data on the induction of ferroptosis, either as monotherapy or in combination with therapies, are limited. For example, sulfasalazine, a system Xc− inhibitor, is currently under investigation in trials targeting solid tumors and glioblastoma (NCT04205357) [177]. Another clinical interventional study is in progress on the use of carbon nanoparticle-loaded iron [CNSI-Fe(II)], which can induce ferroptosis in patients with advanced solid tumors (NCT06048367) [178].

## 4. Conclusions and Future Perspectives

Ferroptosis is a promising cancer therapy strategy that offers a means to overcome treatment resistance and selectively target cancer cells. Chemotherapy combined with ferroptosis inducers enhances therapeutic efficacy, particularly in drug-resistant cancers like ovarian, pancreatic, and colorectal malignancies. These cancers exhibit altered iron metabolism and oxidative stress, making them more susceptible to ferroptosis. Ferroptosis inducers, such as erastin (xCT inhibitor) and RSL3 (GPX4 inhibitor), work synergistically with chemotherapy by increasing lipid peroxidation and intracellular iron levels. Additionally, ferroptosis enhances immune responses by exposing tumor antigens and improving immunotherapy outcomes.

Despite its potential, there are still challenges to be addressed. For example, erastin has poor water solubility and undergoes rapid metabolism, which limits its clinical success. Further research is needed to optimize delivery systems and explore the epigenetic regulation of ferroptosis pathways.

Future studies should investigate new therapeutic approaches, including biomimetic-engineered bacterial conjugates and magnetotactic bacteria as targeted ferroptosis inducers.

Identifying ferroptosis biomarkers is crucial for personalized medicine, enabling tailored treatments that combine ferroptosis inducers with chemotherapy to achieve maximum efficacy and minimal side effects.

AI and machine learning can accelerate drug discovery and combinatorial therapy design, helping overcome multidrug resistance. Integrating ferroptosis into cancer treatment could revolutionize precision oncology and improve patient outcomes.

In the future, better-engineered nano-carriers and nanodrug delivery systems could be employed to improve the effectiveness of ferroptosis inducers.

## Figures and Tables

**Figure 1 ijms-26-03852-f001:**
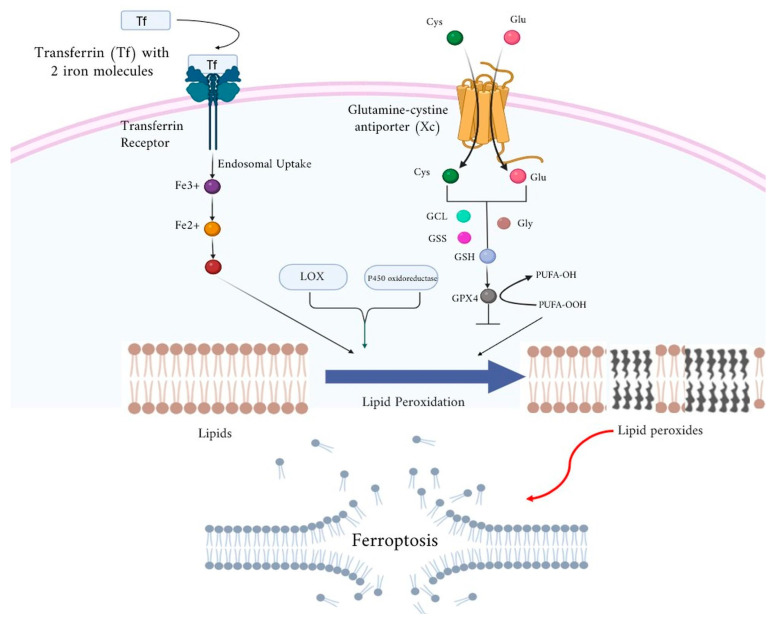
A flow model showing the steps in ferroptosis involving various membrane and intracellular components, including transferrin receptors, glutamine-cystine antiporter (Xc), LOX, GPX, GSH, iron, and lipid peroxides [83].

**Figure 2 ijms-26-03852-f002:**
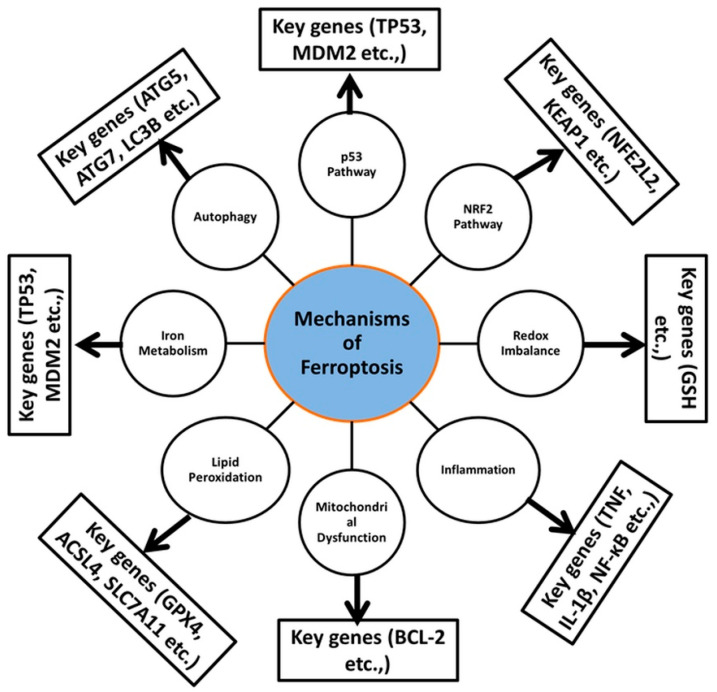
A summary of the various molecular mechanisms of ferroptosis, with key genes involved in the induction of ferroptotic cell death in cancer cells.

**Figure 3 ijms-26-03852-f003:**
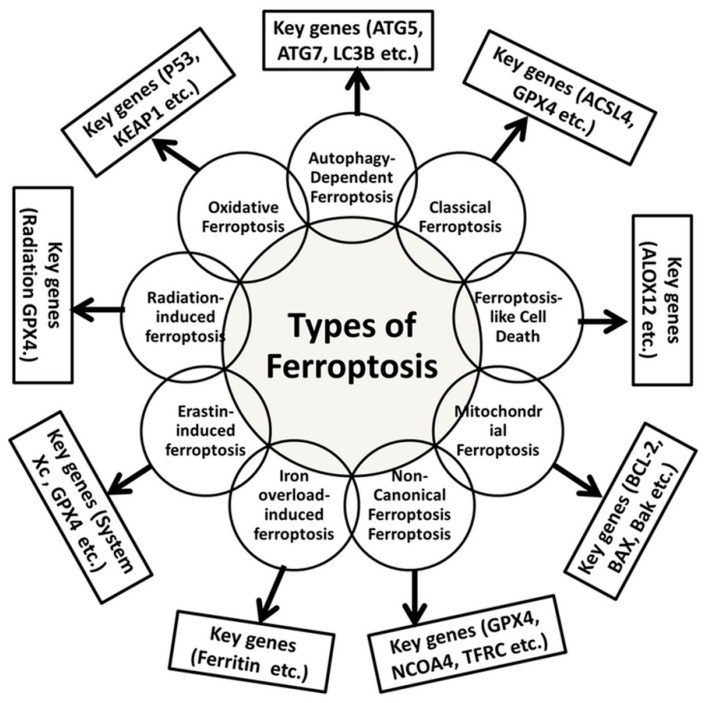
A summary of various types of ferroptosis and their key genes that contribute to ferroptosis cell death [121,122,123,124].

**Table 1 ijms-26-03852-t001:** Various mechanisms of ferroptosis involve molecular genes, proteins, and therapeutic implications for cancer.

Mechanism	Molecular Genes/Proteins	Pathways	Therapeutic Implications	References
Autophagy	ATG5, ATG7, LC3B	Autophagic degradation	Autophagy inhibition	[38]
p53 Pathway	TP53, MDM2	Tumor suppressor regulation	p53-reactivating therapies	[45]
NRF2 Pathway	NFE2L2, KEAP1	Antioxidant response element (ARE) regulation	NRF2 activation	[113]
Redox Imbalance	GSH	Reactive oxygen species (ROS) regulation	Antioxidant therapies	[123]
Inflammation	TNF, IL-1β, NF-κB	Cytokine signaling	Anti-inflammatory therapies	[130,131]
Mitochondrial Dysfunction	BCL-2	Mitochondrial outer membrane permeabilization	Mitochondria-targeting therapies	[132,133]
Lipid Peroxidation	GPX4, ACSL4, SLC7A11	Fatty acid synthesis, antioxidant defenses	Targeting lipid metabolism	[134,135,136,137,138]
Iron Metabolism	NCOA4, TFRC, SLC40A1	Ferritinophagy, iron uptake	Iron chelation therapy	[139,140,141]

**Table 2 ijms-26-03852-t002:** Various types of ferroptosis with molecular descriptions, involved molecular genes, and associated cancer hallmarks.

Types	Molecular Mechanisms	Key Genes Involved	Relevant Cancer Hallmarks	References
Oxidative Ferroptosis	Accumulation of oxygen species (ROS) and mitochondrial damage	P53, KEAP1, NFE2L2	Cancer cell sensitivity	[100]
Autophagy-Dependent Ferroptosis	Involves autophagic degradation of damaged cellular components	ATG5, ATG7, LC3B	Cancer cell survival	[121]
Classical Ferroptosis	Regulated cell death driven by iron-dependent lipid peroxidation	ACSL4, GPX4, SLC7A11	Cancer cell vulnerability	[124,125]
Ferroptosis-like Cell Death	Shares features with ferroptosis but lacks lipid peroxidation	ALOX12	Cancer therapy potential	[127]
Mitochondrial Ferroptosis	Mitochondrial dysfunction, including changes in membrane potential	BCL-2, BAX, Bak	Cancer therapy target	[133]
Non-Canonical Ferroptosis	Independent of GPX4	NCOA4, TFRC, SLC40A1	Resistance in cancer cells	[135,136]
Iron Overload-Induced Ferroptosis	Fenton reaction and lipid peroxidation	Ferritin	Resistance in cancer cells	[141]
Erastin-Induced Ferroptosis	System Xc−, an antiporter of cystine/glutamate	GPX4	Cancer therapy target	[144,145]
Radiation-Induced Ferroptosis	Radiation leads to the generation of ROS	GPX4	Cancer cell sensitivity	[147,148]

**Table 3 ijms-26-03852-t003:** Preclinical and clinical evidence on modulators of ferroptosis in combinatorial anticancer drug approaches, with details on various molecular targets.

Inducers of Ferroptosis	Molecular Target/ Pathway	Combinatorial Anticancer Drug Approach	Preclinical/Clinical Evidence	References
Selenite	GPX4/SELENOP	Anticancer drug	Enhanced antitumor effect in ovarian cancer via induction of ferroptosis and inhibition of GPX4-mediated antioxidant defenses	[143]
PRLX93936 inhibitor of GPX4	GPX4	Cisplatin	upregulation of ROS, lipid peroxidation, and Fe^2+^	[165]
FINO2	GPX4/SLC7A11	Anticancer drug	FINO2 initiates ferroptosis through GPX4 inactivation and iron oxidation	[166]
ML162, ML210	Selenoprotein, TXNRD1	Anticancer drug	Inhibition of selenoprotein, TXNRD1	[169]
Lentivirus sh-GPX4	GPX4	Gefitinib	Increased antitumor efficacy in breast cancer via inhibition of GPX4-mediated ferroptosis and induction of SELENOP-mediated selenium depletion	[171]
Tripterygium glycosides	GPX4/NRF2	Cisplatin	Synergistic antitumor effect in ovarian cancer via induction of ferroptosis and inhibition of NRF2-mediated GPX4 expression	[172]
ML385	GPX4/NRF2	Anticancer drugs	Enhanced antitumor efficacy in ovarian cancer via inhibition of NRF2-mediated GPX4 expression and induction of ferroptosis	[173]
GPX4 inhibitor (RSL3)	GPX4/NRF2	Sorafenib	Synergistic antitumor effect in HCC via inhibition of NRF2-mediated GPX4 expression and induction of ferroptosis	[175]
FINO2	GPX4/SLC7A11	Paclitaxel	Increased antitumor efficacy in breast cancer via inhibition of GPX4-mediated ferroptosis and induction of SLC7A11-mediated glutathione depletion	[176]

## Data Availability

The authors declare that no data generated and submitted and hence not applicable.

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
