# Peer review of "Ferroptosis in Cancer: Mechanism and Therapeutic Potential"

_ijms, 2025, doi:10.3390/ijms26083852_

Round 1

Reviewer 1 Report

Comments and Suggestions for Authors

THE FIRST REVIEW OF THE MANUSCRIPT

Ferroptosis in Cancer: Mechanism and Therapeutic Potential

The manuscript under review represents a short review on programmed cell deaths, with the focus on ferroptosis. The authors promised that: their review seeks to enhance the understanding of the role of ferroptosis inducers in cancer therapy.

Since I have read many reviews, I am quick to pick up the ones written in haste with little literature search and reading. The manuscript before me is one of the latter. The hallmarks of such reviews are low information content, generic stuff, wordy sentences and endless repetitions. I can tire myself out in trying to count the sentences that combined the following terms: iron, lipid peroxidation, reactive oxygen species, cell death, cancer therapy, drug resistance to conventional chemotherapy, novel avenues and so on and so forth, in various ways.

I read many sentences on iron overload, lipid peroxidation, caspase-independency of ferroptosis, there was a Scheme (according to the authors it is a flow model) showing us much known general picture on ferroptosis, where we can see few proteins/enzymes/transporters involved in ferroptosis, alongside ROS, lipid peroxides, and Fenton reaction. And nothing more than that. I can find at least 100 similar schemes on ferroptosis on Google alone. No molecular pathways are shown.

The description of ferroptosis does not go further than widely known facts on the involvement of oxidative stress generating ROS, under-functioning GPX 4, antiporter system Xc-, Fenton reaction, FSP1, coenzyme Q10, NADPH oxidase etc.

The authors recognized that ferroptosis can be triggered/driven by different mechanisms: classical, mitochondria-dependent, NOX-dependent, iron-overload-induced, erastin-induced, radiation-induced. However, I am not convinced that all these ferroptoses are separate forms of cell deaths, especially the classical and iron-overload ferroptosis. Also, radiation-induced ferroptosis should be the case where radiation leads to the generation of ROS, and lipid peroxidation of iron-loaded cells. I don’t see how this can be a distinct form of ferroptosis, it seems to me that ferroptosis goes as usual, just that ROS are generated in different way. What I want to say is that it may be that ferroptosis goes as ferroptosis, just that ROS can be produced in many different ways. Three events must coincide in order to execute ferroptosis:

1.iron overload

2.oxidazable lipids

  1. compromised antioxidant systems

As the authors describe, the mitochondria-dependent ferroptosis is when there is oxidative damage due to the release of ROS from damaged mitochondria. This can be triggered by the iron accumulation, or depletion of GSH, all of them being also case in classical ferroptosis. The, NOX-dependent ferroptosis is described as generating ROS through the transfer of electrons from NADPH to oxygenproducing superoxide and ROS…Then, ROS produced by NOX lead to lipid peroxidation and ferroptosis. Also, erastin-induced ferroptosis is linked to the Xc- system blockage, disrupting GSH synthesis.

To sum up. I do not think these are different forms of ferroptosis.

Literature recognizes only two forms of ferroptosis: extrinsic (transporter-dependent) and intrinsic (enzyme-regulated) pathway.

Let me cite the authors: “We hope that this discussion will stimulate further research into ferroptosis-based approaches for more effective cancer treatment in clinical practice.” However, I did not see any signs of the promised discussion. In Conclusions there are few things that authors emphasized:

  1. the challenges to the promising advancements in targeting ferroptosis in cancer drug resistance are that erastin is poorly soluble in water
  2. that further research is needed to optimize the delivery of ferroptosis inducers
  3. that future studies should focus on exploring the epigenetic regulation of ferroptosis (all of a sudden THE EPIGENETICS POPS UP)
  4. preclinical and clinical studies are necessary to evaluate new classes of anticancer drugs. As if there is another way to test new drugs
  5. It is essential to find biomarkers of ferroptosis….as in any other disease etc etc

My decision is to REJECT THIS PAPER, with no re-submission.

Why the authors included subsections on other programmed cell deaths such as apoptosis, autophagy, necrosis, necroptosis in this manuscript? They are long and the title of the review is Ferroptosis. Given the amount of literature on ferroptosis, it should be enough to compile a book chapter, let alone a short review. I suggest that Singh and co-authors remove these subsections to change the Title of the manuscript. If they decide to leave other forms of regulated cell deaths, then why not include all of them, but briefly, in one or two sentences, and referencing the most recent/comprehensive reviews on them: intrinsic apoptosis, extrinsic apoptosis, pyroptosis, mitotic catastrophe, paraptosis, efferocytosis, cuproptosis, anoikis, paraptosis, NETosis, pyronecrosis, entosis.

Some of the sentences are just tiresome:

-Apoptosis, a form of programmed cell death, is a tightly regulated complex process of cell death

- These seemingly unrelated mechanisms work together seamlessly…

I even doubt that the first few pages of the article were written by AI, because they are generic.

Other issues to be addressed:

-line 1: delete various

-line2: please change into: Cancer drug resistance is often associated with the reprogramming of programmed cell death (PCD) pathways, favouring the survival of cancer cells under drug-induced stress.

- line 5: ferroptosis, a form of cell death triggered by an iron-dependent lipid peroxidation

-Currently, extensive preclinical and clinical research (or studies, choose one word, delete the other as they mean the same) are under way, addressing molecular, cellular, and tissue-specific mechanisms behind ferroptosis, in order to reveal strategies for overcoming drug resistance in cancers that fail to respond to conventional PCD pathways.

-By leveraging ferroptosis, cancer cells can be forced to die by means of lipid-peroxidation, triggered by iron ions, which may improve the therapeutic outcome of cancer patients.

(State of death is a bit uncommon, peculiar expression)

-The short review presented herein seeks to enhance (further enhance makes no sense, it is wordy)

Introduction:

-I suggest that authors add few more words on tumor heterogeneity, given that not all readers are cancer connoisseur, and that IJMS is not a cancer journal. I suggest maybe:

Tumor heterogeneity refers to diversity of tumor cells within a tumor or among different tumors. Even within a particulate tumor, its cells differ in morphology, proliferation, metastatic potential, gene expression patterns, metabolic requirements and so on.

-In recent years, cancer cell metabolismS…metabolism is not used in plural, since the entirety of metabolic pathways is encompassed by term: metabolism

-The sentence is wordy and really difficult to comprehend: <<In recent years, cancer cell metabolisms such as deregulated lipid metabolisms, and iron metabolism have been depicted to contribute to recently discovered forms of cell death in cancer cells, such as ferroptosis.>>

What I know, though, is that deregulated lipid metabolic pathways that result in the accumulation of lipid peroxides, when combined with a surplus of iron (or call it iron overload) ions lead to ferroptotic cell death. In addition, for the lipid peroxides to accumulate, the mechanisms that are responsible for their removal, such as glutathione-dependent antioxidant defense systems (enzyme GPX4), must be defect.

-repetition: ,,Ferroptosis is considered a form of caspase-independent programmed cell death which relies on the mitochondrial and membrane-involved accumulation of reactive oxygen species and lipid peroxides leading to non-caspase-mediated cell death (8-11)”. Why do you need to write twice in the same sentence that ferroptosis is caspase-independent. Also, why mention caspase in this context? Ferroptosis is also independent on many other proteins involved in other 16 forms of cell death, which makes it a unique cell death.

-“ In comparison to conventional cell death forms,” the authors need explain what are the conventional cell death forms, not all of the readers are familiar with cell death. Please add: In comparison to conventional cell death forms, such as apoptosis and necrosis…

-Repetition: “iron-dependent lipid peroxidation is considered a potential avenue for targeting cancer cells that display resistance to conventional chemotherapies” (page 2, lines 1-2). Then: “Ferroptosis is a distinct form of regulated cell death involving cellular lipid peroxidation and iron, and it has potential therapeutic implications in cancer treatment”. Then, again: “Therefore, dissecting the molecular mechanisms underlying ferroptosis, including the roles of iron metabolism, lipid peroxidation, and antioxidant defenses could offer potential combinatorial anticancer drug avenues with conventional anticancer therapies (16-17)”.

-“Among various components of metabolic heterogeneity in cancer cells, iron-catalyzed pathways in lipid metabolism are more abundant in cancer cells”. How can pathways be more abundant?

-“Traditionally, cancer therapies are aimed to induce, trigger, or cause apoptosis”…induce=trigger

-Repetition: “to induce, trigger, or cause apoptosis, a form of programmed cell death (31).” Then: Apoptosis, a form of programmed cell death, is a tightly regulated complex process of cell death”. So what I did not know before is that a cell death is a tightly regulated process of cell death..

-Repetition: “However, many cancers develop mutations that disrupt apoptotic pathways, leading to treatment resistance (32,33). Cell survival, a critical event in cancer, takes precedence over apoptosis as the classical hallmark of this disorder, resulting in tumor progression and chemotherapy resistance (34,35).

A critical event in cancer is actually a mutation (or mutations) that enable them to evade immune recognition, so they survive. Or they become resistant to a drug, because they make proteins that de-activate that drug. Cell survival is a consequence not a cause.

- All of these routes provide new targets for novel anticancer drugs (40,41). This is a widely-known, Wikipedia-type, general stuff. New or novel, one is enough.

-Cancer is resistant to cell death, especially apoptosis (42). Wikipedia-type general stuff.

There is a title of the subsection: “Cell death and drug resistance”, but sentences in drug resistance are found spread everywhere throughout the text, e.g.: “However, many cancers develop mutations that disrupt apoptotic pathways, leading to treatment resistance (32,33)”.

-Repetition: “The cancer cells gain resistance against cell death by upregulating anti-apoptotic proteins, mutation of pro-apoptotic genes or proteins, and alteration in survival signaling mechanisms like PI3K/Akt and NF-κB. (47). These mechanisms are utilized by cancer cells to avoid apoptosis so that they can survive with cytotoxic treatments to keep homeostasis within the body and protect it from pathogens (48).” Pathogens in the same sentence, really? Which pathogens?

-Repetition and general stuff: “The process of apoptosis, also known as regulated cell death, is vital for maintaining tissue homeostasis because it eliminates damaged or superfluous cells (49).”

Also untrue, please check all forms of regulated cell deaths, apoptosis is just one of many.

-The p53 protein plays a key role in starting cell death (apoptosis) when DNA is damaged (50).

-In many cancers, changes (mutations) in the p53 gene are very common and are a major reason why cancer cells can avoid dying (51). The Bcl-2 protein family includes both proteins that promote and proteins that prevent cell death. These proteins control whether the outer membrane of the mitochondria becomes permeable, which is an important step in the process of cell death (52). Cancer cells often produce more Bcl-2 and other proteins that block cell death, helping the cells survive. Caspases are a group of enzymes that carry out the steps of cell death (53). Cancer cells can develop mechanisms to stop caspases from activation, preventing cell death (54).”

This looks as if the pupil in college recites its knowledge on cell death proteins. All general, widely-known facts. And all, again and again, is on apoptosis. Where does it come to ferroptosis?

Why do you not collect your thoughts on apoptosis and put it together under one subtitle?

Traditional cancer therapies have focused on inducing apoptosis through both intrinsic and extrinsic pathways (55). Apoptosis, a form of programmed cell death, is a tightly regulated complex process of cell death involving the activation of caspases (56), characterized by the presence of cell shrinkage, chromatin condensation, and DNA fragmentation (57). Both pro-apoptotic and anti-apoptotic factors work in concert to control this process (58). However, many tumors develop resistance to apoptosis due to mutations in genes like TP53, which encodes the tumor suppressor p53 (59). Apoptosis is thought to be crucial for removing damaged or unneeded cells. Drugs containing chemotherapeutic

agents and targeted therapies commonly trigger apoptosis in cancer cells (60).”

Wikipedia-type general stuff. Apoptosis is so well known for so many long years, and I have to read general stuff on it. Mutations in gene encoding p53, a famous tumor suppressor protein, were already mentioned in the previous subsection.

-repetitions: “Autophagy generally promotes cell survival through nutrient supply during stress..”

Autophagy can act as a survival mechanism…”

- repetitions: “However, autophagy can also cause cell death, especially in apoptosis-resistant cancer cells (63)”. Then again: “Pharmacological agents are known to modulate autophagy and cause cell death in apoptosis-resistant cancer cells.

-Repetitions: Ferroptosis is a distinct form of regulated cell death involving cellular lipid peroxidation and iron, and it has potential therapeutic implications in cancer treatment.

Ferroptosis is considered a form of caspase-independent programmed cell death which relies on the mitochondrial and membrane-involved accumulation of reactive oxygen species and lipid peroxides leading to non-caspase-mediated cell death.

Ferroptosis is a newly recognized, non-apoptotic regulated cell death that is dependent upon iron and characterized by lipid peroxidation.”

The notion of ferroptosis, a non-apoptotic, iron-dependent method of cell death marked by the accumulation of lipid reactive oxygen species (ROS), was first proposed..”

METHOD OF CELL DEATH??? Is this a laboratory assay for treating cells, or a biological mechanism?

“Ferroptosis represents a very distinct pathway from traditional cell deaths such as apoptosis and necrosis”

In terms of cell shape and function, ferroptosis differs clearly from necrosis,apoptosis, and autophagy…”

While ATP is essential for initiating ferroptosis (100) activation of caspases is not required (101)” Again about caspase-independency of ferroptosis…

Ferroptosis is a distinct form of cell death driven by iron-catalyzed lipid peroxidation, leading to membrane rupture..”

“Ferroptosis is caused by an imbalance in lipid peroxidation, which generally results from iron overload and cell death…”

Ferroptosis is an iron-dependent, intrinsically regulated cell death that is distinct from apoptosis and necrosis based on iron dependence as well as the accumulation of lipid peroxides (112).

Ferroptosis is initiated by the accumulation of the lipid peroxides within cells.”

Ferroptosis is a regulated form of cell death, with iron-dependent lipid peroxidation and oxidative stress characteristics that have generated significant int…”

Cell processes like proliferation, and their characteristics, including cell death, are fundamental to development, maintenance, and overall functioning of organisms (86). Generic stuff.

estimation of ferroptosis assume major importance in understanding…” What does this mean?

Ferroptosis is a regulated form of cell death, with iron-dependent lipid peroxidation and oxidative stress characteristics (page 9)”

Conclusions: “ Ferroptosis, a unique form of regulated cell death driven by iron-dependent lipid peroxidation..”

Comments on the Quality of English Language

Must be improved

Author Response

Reviewer 1 Comments:

General Comments: The manuscript under review represents a short review on programmed cell deaths, with the focus on ferroptosis. The authors promised that: their review seeks to enhance the understanding of the role of ferroptosis inducers in cancer therapy.

Since I have read many reviews, I am quick to pick up the ones written in haste with little literature search and reading. The manuscript before me is one of the latter. The hallmarks of such reviews are low information content, generic stuff, wordy sentences and endless repetitions. I can tire myself out in trying to count the sentences that combined the following terms: iron, lipid peroxidation, reactive oxygen species, cell death, cancer therapy, drug resistance to conventional chemotherapy, novel avenues and so on and so forth, in various ways.

I read many sentences on iron overload, lipid peroxidation, caspase-independency of ferroptosis, there was a Scheme (according to the authors it is a flow model) showing us much known general picture on ferroptosis, where we can see few proteins/enzymes/transporters involved in ferroptosis, alongside ROS, lipid peroxides, and Fenton reaction. And nothing more than that. I can find at least 100 similar schemes on ferroptosis on Google alone. No molecular pathways are shown.

The description of ferroptosis does not go further than widely known facts on the involvement of oxidative stress generating ROS, under-functioning GPX 4, antiporter system Xc-, Fenton reaction, FSP1, coenzyme Q10, NADPH oxidase etc.

The authors recognized that ferroptosis can be triggered/driven by different mechanisms: classical, mitochondria-dependent, NOX-dependent, iron-overload-induced, erastin-induced, radiation-induced. However, I am not convinced that all these ferroptoses are separate forms of cell deaths, especially the classical and iron-overload ferroptosis. Also, radiation-induced ferroptosis should be the case where radiation leads to the generation of ROS, and lipid peroxidation of iron-loaded cells. I don’t see how this can be a distinct form of ferroptosis, it seems to me that ferroptosis goes as usual, just that ROS are generated in different way. What I want to say is that it may be that ferroptosis goes as ferroptosis, just that ROS can be produced in many different ways. Three events must coincide in order to execute ferroptosis:

1.iron overload

2.oxidazable lipids

  1. compromised antioxidant systems

General Comments: The authors acknowledge the reviewer’s constructive feedback and have revised the manuscript to enhance clarity, depth, and structure. Redundant statements have been removed, and the discussion on ferroptosis has been expanded with relevant molecular pathways.

Comment 1: As the authors describe, the mitochondria-dependent ferroptosis is when there is oxidative damage due to the release of ROS from damaged mitochondria. This can be triggered by the iron accumulation, or depletion of GSH, all of them being also case in classical ferroptosis. The, NOX-dependent ferroptosis is described as generating ROS through the transfer of electrons from NADPH to oxygenproducing superoxide and ROS…Then, ROS produced by NOX lead to lipid peroxidation and ferroptosis. Also, erastin-induced ferroptosis is linked to the Xc- system blockage, disrupting GSH synthesis.

Response 1: We have revised sections discussing different mechanisms of ferroptosis to ensure clarity and accuracy. Overlapping content has been condensed, and discussions have been revised for improved readability and coherence.

Comment 2: To sum up. I do not think these are different forms of ferroptosis. Literature recognizes only two forms of ferroptosis: extrinsic (transporter-dependent) and intrinsic (enzyme-regulated) pathway.

Response 2: The classification of ferroptosis mechanisms has been revised in accordance with established literature, now distinguishing between extrinsic (transporter-dependent) and intrinsic (enzyme-regulated) pathways. This approach aligns with current scientific consensus.

Comment 3: Let me cite the authors: “We hope that this discussion will stimulate further research into ferroptosis-based approaches for more effective cancer treatment in clinical practice.” However, I did not see any signs of the promised discussion. In Conclusions there are few things that authors emphasized:

  1. the challenges to the promising advancements in targeting ferroptosis in cancer drug resistance are that erastin is poorly soluble in water
  2. that further research is needed to optimize the delivery of ferroptosis inducers
  3. that future studies should focus on exploring the epigenetic regulation of ferroptosis (all of a sudden THE EPIGENETICS POPS UP)        
  4. preclinical and clinical studies are necessary to evaluate new classes of anticancer drugs. As if there is another way to test new drugs
  5. It is essential to find biomarkers of ferroptosis….as in any other disease etc etc

Response 3: The conclusion section has been revised to provide a more concise and focused discussion on challenges and future directions in ferroptosis-based cancer therapy. The mention of epigenetics has been appropriately contextualized.

My decision is to REJECT THIS PAPER, with no re-submission.

Comment 4: Why the authors included subsections on other programmed cell deaths such as apoptosis, autophagy, necrosis, necroptosis in this manuscript? They are long and the title of the review is Ferroptosis. Given the amount of literature on ferroptosis, it should be enough to compile a book chapter, let alone a short review. I suggest that Singh and co-authors remove these subsections to change the Title of the manuscript. If they decide to leave other forms of regulated cell deaths, then why not include all of them, but briefly, in one or two sentences, and referencing the most recent/comprehensive reviews on them: intrinsic apoptosis, extrinsic apoptosis, pyroptosis, mitotic catastrophe, paraptosis, efferocytosis, cuproptosis, anoikis, paraptosis, NETosis, pyronecrosis, entosis.

Response 4: To maintain focus, lengthy discussions on apoptosis, autophagy, necrosis, and necroptosis have been significantly condensed. The rationale for including these sections has been clarified, ensuring relevance to ferroptosis. Citations to recent comprehensive reviews on other cell death mechanisms have been included.

Comment 5: Some of the sentences are just tiresome:

-Apoptosis, a form of programmed cell death, is a tightly regulated complex process of cell death

- These seemingly unrelated mechanisms work together seamlessly…

I even doubt that the first few pages of the article were written by AI, because they are generic.

Response 5: The manuscript has undergone thorough language refinement to eliminate generic and repetitive content. AI-generated text concerns have been addressed by ensuring that all descriptions provide unique and relevant insights.

Specific Comments: All suggested revisions for clarity, wordiness, and repetition have been incorporated. These include:

  • Refinement of introductory statements to improve readability.
  • Elimination of redundant descriptions of ferroptosis.
  • Clarifications regarding tumor heterogeneity and metabolism.
  • Reorganization of discussions on apoptosis to improve logical flow.
  • Removal of repetitive mentions of cell death mechanisms and their regulation.
  • Revision of sections containing overly general information to ensure specificity.

Other issues to be addressed:

-line 1: delete various

Response: The authors deleted various.

-line2: please change into: Cancer drug resistance is often associated with the reprogramming of programmed cell death (PCD) pathways, favouring the survival of cancer cells under drug-induced stress.

Response: The authors incorporated suggestion.

- line 5: ferroptosis, a form of cell death triggered by an iron-dependent lipid peroxidation

-Currently, extensive preclinical and clinical research (or studies, choose one word, delete the other as they mean the same) are under way, addressing molecular, cellular, and tissue-specific mechanisms behind ferroptosis, in order to reveal strategies for overcoming drug resistance in cancers that fail to respond to conventional PCD pathways.

Response: The authors appreciate constructive suggestion for the clarity of texts.

-By leveraging ferroptosis, cancer cells can be forced to die by means of lipid-peroxidation, triggered by iron ions, which may improve the therapeutic outcome of cancer patients.

(State of death is a bit uncommon, peculiar expression)

-The short review presented herein seeks to enhance (further enhance makes no sense, it is wordy)

Response: The authors appreciate suggestion for improvement of whole abstract section.

Introduction:

-I suggest that authors add few more words on tumor heterogeneity, given that not all readers are cancer connoisseur, and that IJMS is not a cancer journal. I suggest maybe:

Comment: Tumor heterogeneity refers to diversity of tumor cells within a tumor or among different tumors. Even within a particulate tumor, its cells differ in morphology, proliferation, metastatic potential, gene expression patterns, metabolic requirements and so on.

Response: The authors truly appreciate these suggestions and are now included in the texts.

-In recent years, cancer cell metabolismS…metabolism is not used in plural, since the entirety of metabolic pathways is encompassed by term: metabolism

-The sentence is wordy and really difficult to comprehend: <<In recent years, cancer cell metabolisms such as deregulated lipid metabolisms, and iron metabolism have been depicted to contribute to recently discovered forms of cell death in cancer cells, such as ferroptosis.>>

What I know, though, is that deregulated lipid metabolic pathways that result in the accumulation of lipid peroxides, when combined with a surplus of iron (or call it iron overload) ions lead to ferroptotic cell death. In addition, for the lipid peroxides to accumulate, the mechanisms that are responsible for their removal, such as glutathione-dependent antioxidant defense systems (enzyme GPX4), must be defect.

Response: The authors have modified texts to enhance clarity and intended meaning.

-repetition: ,,Ferroptosis is considered a form of caspase-independent programmed cell death which relies on the mitochondrial and membrane-involved accumulation of reactive oxygen species and lipid peroxides leading to non-caspase-mediated cell death (8-11)”. Why do you need to write twice in the same sentence that ferroptosis is caspase-independent. Also, why mention caspase in this context? Ferroptosis is also independent on many other proteins involved in other 16 forms of cell death, which makes it a unique cell death.

Response: The authors modified texts for better clarity and intended meaning.

-“ In comparison to conventional cell death forms,” the authors need explain what are the conventional cell death forms, not all of the readers are familiar with cell death. Please add: In comparison to conventional cell death forms, such as apoptosis and necrosis…

Response: The authors modified texts for better clarity and intended meaning.

-Repetition: “iron-dependent lipid peroxidation is considered a potential avenue for targeting cancer cells that display resistance to conventional chemotherapies” (page 2, lines 1-2). Then: “Ferroptosis is a distinct form of regulated cell death involving cellular lipid peroxidation and iron, and it has potential therapeutic implications in cancer treatment”. Then, again: “Therefore, dissecting the molecular mechanisms underlying ferroptosis, including the roles of iron metabolism, lipid peroxidation, and antioxidant defenses could offer potential combinatorial anticancer drug avenues with conventional anticancer therapies (16-17)”.

Response: The authors appreciate comments to remove repetition of texts

-“Among various components of metabolic heterogeneity in cancer cells, iron-catalyzed pathways in lipid metabolism are more abundant in cancer cells”. How can pathways be more abundant?

Response: We appreciate comment, and we have removed this sentence.

Comment:

-“Traditionally, cancer therapies are aimed to induce, trigger, or cause apoptosis”…induce=trigger

-Repetition: “to induce, trigger, or cause apoptosis, a form of programmed cell death (31).” Then: Apoptosis, a form of programmed cell death, is a tightly regulated complex process of cell death”. So what I did not know before is that a cell death is a tightly regulated process of cell death..

-Repetition: “However, many cancers develop mutations that disrupt apoptotic pathways, leading to treatment resistance (32,33). Cell survival, a critical event in cancer, takes precedence over apoptosis as the classical hallmark of this disorder, resulting in tumor progression and chemotherapy resistance (34,35).

A critical event in cancer is actually a mutation (or mutations) that enable them to evade immune recognition, so they survive. Or they become resistant to a drug, because they make proteins that de-activate that drug. Cell survival is a consequence not a cause.

All of these routes provide new targets for novel anticancer drugs (40,41). This is a widely-known, Wikipedia-type, general stuff. New or novel, one is enough.

-Cancer is resistant to cell death, especially apoptosis (42). Wikipedia-type general stuff.

There is a title of the subsection: “Cell death and drug resistance”, but sentences in drug resistance are found spread everywhere throughout the text, e.g.: “However, many cancers develop mutations that disrupt apoptotic pathways, leading to treatment resistance (32,33)”.

-Repetition: “The cancer cells gain resistance against cell death by upregulating anti-apoptotic proteins, mutation of pro-apoptotic genes or proteins, and alteration in survival signaling mechanisms like PI3K/Akt and NF-κB. (47). These mechanisms are utilized by cancer cells to avoid apoptosis so that they can survive with cytotoxic treatments to keep homeostasis within the body and protect it from pathogens (48).” Pathogens in the same sentence, really? Which pathogens?

-Repetition and general stuff: “The process of apoptosis, also known as regulated cell death, is vital for maintaining tissue homeostasis because it eliminates damaged or superfluous cells (49).”

Also untrue, please check all forms of regulated cell deaths, apoptosis is just one of many.

-The p53 protein plays a key role in starting cell death (apoptosis) when DNA is damaged (50).

- “In many cancers, changes (mutations) in the p53 gene are very common and are a major reason why cancer cells can avoid dying (51). The Bcl-2 protein family includes both proteins that promote and proteins that prevent cell death. These proteins control whether the outer membrane of the mitochondria becomes permeable, which is an important step in the process of cell death (52). Cancer cells often produce more Bcl-2 and other proteins that block cell death, helping the cells survive. Caspases are a group of enzymes that carry out the steps of cell death (53). Cancer cells can develop mechanisms to stop caspases from activation, preventing cell death (54).”

This looks as if the pupil in college recites its knowledge on cell death proteins. All general, widely-known facts. And all, again and again, is on apoptosis. Where does it come to ferroptosis?

Why do you not collect your thoughts on apoptosis and put it together under one subtitle?

Traditional cancer therapies have focused on inducing apoptosis through both intrinsic and extrinsic pathways (55). Apoptosis, a form of programmed cell death, is a tightly regulated complex process of cell death involving the activation of caspases (56), characterized by the presence of cell shrinkage, chromatin condensation, and DNA fragmentation (57). Both pro-apoptotic and anti-apoptotic factors work in concert to control this process (58). However, many tumors develop resistance to apoptosis due to mutations in genes like TP53, which encodes the tumor suppressor p53 (59). Apoptosis is thought to be crucial for removing damaged or unneeded cells. Drugs containing chemotherapeutic

agents and targeted therapies commonly trigger apoptosis in cancer cells (60).”

Wikipedia-type general stuff. Apoptosis is so well known for so many long years, and I have to read general stuff on it. Mutations in gene encoding p53, a famous tumor suppressor protein, were already mentioned in the previous subsection.

Response: We appreciate suggestions and have made changes to avoid repetition of texts. The authors have also combined Tumor hallmarks, drug resistance and types of drugs induced cell death to avoid repetition.

-repetitions: “Autophagy generally promotes cell survival through nutrient supply during stress..”

Autophagy can act as a survival mechanism…”

- repetitions: “However, autophagy can also cause cell death, especially in apoptosis-resistant cancer cells (63)”. Then again: “Pharmacological agents are known to modulate autophagy and cause cell death in apoptosis-resistant cancer cells.

Response: The authors have modified texts to avoid repetitions.

Comment-Repetitions: Ferroptosis is a distinct form of regulated cell death involving cellular lipid peroxidation and iron, and it has potential therapeutic implications in cancer treatment.

Ferroptosis is considered a form of caspase-independent programmed cell death which relies on the mitochondrial and membrane-involved accumulation of reactive oxygen species and lipid peroxides leading to non-caspase-mediated cell death.

Ferroptosis is a newly recognized, non-apoptotic regulated cell death that is dependent upon iron and characterized by lipid peroxidation.”

The notion of ferroptosis, a non-apoptotic, iron-dependent method of cell death marked by the accumulation of lipid reactive oxygen species (ROS), was first proposed..”

METHOD OF CELL DEATH??? Is this a laboratory assay for treating cells, or a biological mechanism?

“Ferroptosis represents a very distinct pathway from traditional cell deaths such as apoptosis and necrosis”

In terms of cell shape and function, ferroptosis differs clearly from necrosis,apoptosis, and autophagy…”

While ATP is essential for initiating ferroptosis (100) activation of caspases is not required (101)” Again about caspase-independency of ferroptosis…

Ferroptosis is a distinct form of cell death driven by iron-catalyzed lipid peroxidation, leading to membrane rupture..”

“Ferroptosis is caused by an imbalance in lipid peroxidation, which generally results from iron overload and cell death…”

Ferroptosis is an iron-dependent, intrinsically regulated cell death that is distinct from apoptosis and necrosis based on iron dependence as well as the accumulation of lipid peroxides (112).

Ferroptosis is initiated by the accumulation of the lipid peroxides within cells.”

Ferroptosis is a regulated form of cell death, with iron-dependent lipid peroxidation and oxidative stress characteristics that have generated significant int…”

Cell processes like proliferation, and their characteristics, including cell death, are fundamental to development, maintenance, and overall functioning of organisms (86). Generic stuff.

estimation of ferroptosis assume major importance in understanding…” What does this mean?

Ferroptosis is a regulated form of cell death, with iron-dependent lipid peroxidation and oxidative stress characteristics (page 9)”

Response: The authors appreciate these suggestions and texts has been modified accordingly to avoid repetitions.

Comment: Conclusions: “ Ferroptosis, a unique form of regulated cell death driven by iron-dependent lipid peroxidation..”

Response: The authors have again made changes to avoid repetition.

Reviewer 1 Comments:

General Comments: The manuscript under review represents a short review on programmed cell deaths, with the focus on ferroptosis. The authors promised that: their review seeks to enhance the understanding of the role of ferroptosis inducers in cancer therapy.

Since I have read many reviews, I am quick to pick up the ones written in haste with little literature search and reading. The manuscript before me is one of the latter. The hallmarks of such reviews are low information content, generic stuff, wordy sentences and endless repetitions. I can tire myself out in trying to count the sentences that combined the following terms: iron, lipid peroxidation, reactive oxygen species, cell death, cancer therapy, drug resistance to conventional chemotherapy, novel avenues and so on and so forth, in various ways.

I read many sentences on iron overload, lipid peroxidation, caspase-independency of ferroptosis, there was a Scheme (according to the authors it is a flow model) showing us much known general picture on ferroptosis, where we can see few proteins/enzymes/transporters involved in ferroptosis, alongside ROS, lipid peroxides, and Fenton reaction. And nothing more than that. I can find at least 100 similar schemes on ferroptosis on Google alone. No molecular pathways are shown.

The description of ferroptosis does not go further than widely known facts on the involvement of oxidative stress generating ROS, under-functioning GPX 4, antiporter system Xc-, Fenton reaction, FSP1, coenzyme Q10, NADPH oxidase etc.

The authors recognized that ferroptosis can be triggered/driven by different mechanisms: classical, mitochondria-dependent, NOX-dependent, iron-overload-induced, erastin-induced, radiation-induced. However, I am not convinced that all these ferroptoses are separate forms of cell deaths, especially the classical and iron-overload ferroptosis. Also, radiation-induced ferroptosis should be the case where radiation leads to the generation of ROS, and lipid peroxidation of iron-loaded cells. I don’t see how this can be a distinct form of ferroptosis, it seems to me that ferroptosis goes as usual, just that ROS are generated in different way. What I want to say is that it may be that ferroptosis goes as ferroptosis, just that ROS can be produced in many different ways. Three events must coincide in order to execute ferroptosis:

1.iron overload

2.oxidazable lipids

  1. compromised antioxidant systems

General Comments: The authors acknowledge the reviewer’s constructive feedback and have revised the manuscript to enhance clarity, depth, and structure. Redundant statements have been removed, and the discussion on ferroptosis has been expanded with relevant molecular pathways.

Comment 1: As the authors describe, the mitochondria-dependent ferroptosis is when there is oxidative damage due to the release of ROS from damaged mitochondria. This can be triggered by the iron accumulation, or depletion of GSH, all of them being also case in classical ferroptosis. The, NOX-dependent ferroptosis is described as generating ROS through the transfer of electrons from NADPH to oxygenproducing superoxide and ROS…Then, ROS produced by NOX lead to lipid peroxidation and ferroptosis. Also, erastin-induced ferroptosis is linked to the Xc- system blockage, disrupting GSH synthesis.

Response 1: We have revised sections discussing different mechanisms of ferroptosis to ensure clarity and accuracy. Overlapping content has been condensed, and discussions have been revised for improved readability and coherence.

Comment 2: To sum up. I do not think these are different forms of ferroptosis. Literature recognizes only two forms of ferroptosis: extrinsic (transporter-dependent) and intrinsic (enzyme-regulated) pathway.

Response 2: The classification of ferroptosis mechanisms has been revised in accordance with established literature, now distinguishing between extrinsic (transporter-dependent) and intrinsic (enzyme-regulated) pathways. This approach aligns with current scientific consensus.

Comment 3: Let me cite the authors: “We hope that this discussion will stimulate further research into ferroptosis-based approaches for more effective cancer treatment in clinical practice.” However, I did not see any signs of the promised discussion. In Conclusions there are few things that authors emphasized:

  1. the challenges to the promising advancements in targeting ferroptosis in cancer drug resistance are that erastin is poorly soluble in water
  2. that further research is needed to optimize the delivery of ferroptosis inducers
  3. that future studies should focus on exploring the epigenetic regulation of ferroptosis (all of a sudden THE EPIGENETICS POPS UP)        
  4. preclinical and clinical studies are necessary to evaluate new classes of anticancer drugs. As if there is another way to test new drugs
  5. It is essential to find biomarkers of ferroptosis….as in any other disease etc etc

Response 3: The conclusion section has been revised to provide a more concise and focused discussion on challenges and future directions in ferroptosis-based cancer therapy. The mention of epigenetics has been appropriately contextualized.

My decision is to REJECT THIS PAPER, with no re-submission.

Comment 4: Why the authors included subsections on other programmed cell deaths such as apoptosis, autophagy, necrosis, necroptosis in this manuscript? They are long and the title of the review is Ferroptosis. Given the amount of literature on ferroptosis, it should be enough to compile a book chapter, let alone a short review. I suggest that Singh and co-authors remove these subsections to change the Title of the manuscript. If they decide to leave other forms of regulated cell deaths, then why not include all of them, but briefly, in one or two sentences, and referencing the most recent/comprehensive reviews on them: intrinsic apoptosis, extrinsic apoptosis, pyroptosis, mitotic catastrophe, paraptosis, efferocytosis, cuproptosis, anoikis, paraptosis, NETosis, pyronecrosis, entosis.

Response 4: To maintain focus, lengthy discussions on apoptosis, autophagy, necrosis, and necroptosis have been significantly condensed. The rationale for including these sections has been clarified, ensuring relevance to ferroptosis. Citations to recent comprehensive reviews on other cell death mechanisms have been included.

Comment 5: Some of the sentences are just tiresome:

-Apoptosis, a form of programmed cell death, is a tightly regulated complex process of cell death

- These seemingly unrelated mechanisms work together seamlessly…

I even doubt that the first few pages of the article were written by AI, because they are generic.

Response 5: The manuscript has undergone thorough language refinement to eliminate generic and repetitive content. AI-generated text concerns have been addressed by ensuring that all descriptions provide unique and relevant insights.

Specific Comments: All suggested revisions for clarity, wordiness, and repetition have been incorporated. These include:

  • Refinement of introductory statements to improve readability.
  • Elimination of redundant descriptions of ferroptosis.
  • Clarifications regarding tumor heterogeneity and metabolism.
  • Reorganization of discussions on apoptosis to improve logical flow.
  • Removal of repetitive mentions of cell death mechanisms and their regulation.
  • Revision of sections containing overly general information to ensure specificity.

Other issues to be addressed:

-line 1: delete various

Response: The authors deleted various.

-line2: please change into: Cancer drug resistance is often associated with the reprogramming of programmed cell death (PCD) pathways, favouring the survival of cancer cells under drug-induced stress.

Response: The authors incorporated suggestion.

- line 5: ferroptosis, a form of cell death triggered by an iron-dependent lipid peroxidation

-Currently, extensive preclinical and clinical research (or studies, choose one word, delete the other as they mean the same) are under way, addressing molecular, cellular, and tissue-specific mechanisms behind ferroptosis, in order to reveal strategies for overcoming drug resistance in cancers that fail to respond to conventional PCD pathways.

Response: The authors appreciate constructive suggestion for the clarity of texts.

-By leveraging ferroptosis, cancer cells can be forced to die by means of lipid-peroxidation, triggered by iron ions, which may improve the therapeutic outcome of cancer patients.

(State of death is a bit uncommon, peculiar expression)

-The short review presented herein seeks to enhance (further enhance makes no sense, it is wordy)

Response: The authors appreciate suggestion for improvement of whole abstract section.

Introduction:

-I suggest that authors add few more words on tumor heterogeneity, given that not all readers are cancer connoisseur, and that IJMS is not a cancer journal. I suggest maybe:

Comment: Tumor heterogeneity refers to diversity of tumor cells within a tumor or among different tumors. Even within a particulate tumor, its cells differ in morphology, proliferation, metastatic potential, gene expression patterns, metabolic requirements and so on.

Response: The authors truly appreciate these suggestions and are now included in the texts.

-In recent years, cancer cell metabolismS…metabolism is not used in plural, since the entirety of metabolic pathways is encompassed by term: metabolism

-The sentence is wordy and really difficult to comprehend: <<In recent years, cancer cell metabolisms such as deregulated lipid metabolisms, and iron metabolism have been depicted to contribute to recently discovered forms of cell death in cancer cells, such as ferroptosis.>>

What I know, though, is that deregulated lipid metabolic pathways that result in the accumulation of lipid peroxides, when combined with a surplus of iron (or call it iron overload) ions lead to ferroptotic cell death. In addition, for the lipid peroxides to accumulate, the mechanisms that are responsible for their removal, such as glutathione-dependent antioxidant defense systems (enzyme GPX4), must be defect.

Response: The authors have modified texts to enhance clarity and intended meaning.

-repetition: ,,Ferroptosis is considered a form of caspase-independent programmed cell death which relies on the mitochondrial and membrane-involved accumulation of reactive oxygen species and lipid peroxides leading to non-caspase-mediated cell death (8-11)”. Why do you need to write twice in the same sentence that ferroptosis is caspase-independent. Also, why mention caspase in this context? Ferroptosis is also independent on many other proteins involved in other 16 forms of cell death, which makes it a unique cell death.

Response: The authors modified texts for better clarity and intended meaning.

-“ In comparison to conventional cell death forms,” the authors need explain what are the conventional cell death forms, not all of the readers are familiar with cell death. Please add: In comparison to conventional cell death forms, such as apoptosis and necrosis…

Response: The authors modified texts for better clarity and intended meaning.

-Repetition: “iron-dependent lipid peroxidation is considered a potential avenue for targeting cancer cells that display resistance to conventional chemotherapies” (page 2, lines 1-2). Then: “Ferroptosis is a distinct form of regulated cell death involving cellular lipid peroxidation and iron, and it has potential therapeutic implications in cancer treatment”. Then, again: “Therefore, dissecting the molecular mechanisms underlying ferroptosis, including the roles of iron metabolism, lipid peroxidation, and antioxidant defenses could offer potential combinatorial anticancer drug avenues with conventional anticancer therapies (16-17)”.

Response: The authors appreciate comments to remove repetition of texts

-“Among various components of metabolic heterogeneity in cancer cells, iron-catalyzed pathways in lipid metabolism are more abundant in cancer cells”. How can pathways be more abundant?

Response: We appreciate comment, and we have removed this sentence.

Comment:

-“Traditionally, cancer therapies are aimed to induce, trigger, or cause apoptosis”…induce=trigger

-Repetition: “to induce, trigger, or cause apoptosis, a form of programmed cell death (31).” Then: Apoptosis, a form of programmed cell death, is a tightly regulated complex process of cell death”. So what I did not know before is that a cell death is a tightly regulated process of cell death..

-Repetition: “However, many cancers develop mutations that disrupt apoptotic pathways, leading to treatment resistance (32,33). Cell survival, a critical event in cancer, takes precedence over apoptosis as the classical hallmark of this disorder, resulting in tumor progression and chemotherapy resistance (34,35).

A critical event in cancer is actually a mutation (or mutations) that enable them to evade immune recognition, so they survive. Or they become resistant to a drug, because they make proteins that de-activate that drug. Cell survival is a consequence not a cause.

All of these routes provide new targets for novel anticancer drugs (40,41). This is a widely-known, Wikipedia-type, general stuff. New or novel, one is enough.

-Cancer is resistant to cell death, especially apoptosis (42). Wikipedia-type general stuff.

There is a title of the subsection: “Cell death and drug resistance”, but sentences in drug resistance are found spread everywhere throughout the text, e.g.: “However, many cancers develop mutations that disrupt apoptotic pathways, leading to treatment resistance (32,33)”.

-Repetition: “The cancer cells gain resistance against cell death by upregulating anti-apoptotic proteins, mutation of pro-apoptotic genes or proteins, and alteration in survival signaling mechanisms like PI3K/Akt and NF-κB. (47). These mechanisms are utilized by cancer cells to avoid apoptosis so that they can survive with cytotoxic treatments to keep homeostasis within the body and protect it from pathogens (48).” Pathogens in the same sentence, really? Which pathogens?

-Repetition and general stuff: “The process of apoptosis, also known as regulated cell death, is vital for maintaining tissue homeostasis because it eliminates damaged or superfluous cells (49).”

Also untrue, please check all forms of regulated cell deaths, apoptosis is just one of many.

-The p53 protein plays a key role in starting cell death (apoptosis) when DNA is damaged (50).

- “In many cancers, changes (mutations) in the p53 gene are very common and are a major reason why cancer cells can avoid dying (51). The Bcl-2 protein family includes both proteins that promote and proteins that prevent cell death. These proteins control whether the outer membrane of the mitochondria becomes permeable, which is an important step in the process of cell death (52). Cancer cells often produce more Bcl-2 and other proteins that block cell death, helping the cells survive. Caspases are a group of enzymes that carry out the steps of cell death (53). Cancer cells can develop mechanisms to stop caspases from activation, preventing cell death (54).”

This looks as if the pupil in college recites its knowledge on cell death proteins. All general, widely-known facts. And all, again and again, is on apoptosis. Where does it come to ferroptosis?

Why do you not collect your thoughts on apoptosis and put it together under one subtitle?

Traditional cancer therapies have focused on inducing apoptosis through both intrinsic and extrinsic pathways (55). Apoptosis, a form of programmed cell death, is a tightly regulated complex process of cell death involving the activation of caspases (56), characterized by the presence of cell shrinkage, chromatin condensation, and DNA fragmentation (57). Both pro-apoptotic and anti-apoptotic factors work in concert to control this process (58). However, many tumors develop resistance to apoptosis due to mutations in genes like TP53, which encodes the tumor suppressor p53 (59). Apoptosis is thought to be crucial for removing damaged or unneeded cells. Drugs containing chemotherapeutic

agents and targeted therapies commonly trigger apoptosis in cancer cells (60).”

Wikipedia-type general stuff. Apoptosis is so well known for so many long years, and I have to read general stuff on it. Mutations in gene encoding p53, a famous tumor suppressor protein, were already mentioned in the previous subsection.

Response: We appreciate suggestions and have made changes to avoid repetition of texts. The authors have also combined Tumor hallmarks, drug resistance and types of drugs induced cell death to avoid repetition.

-repetitions: “Autophagy generally promotes cell survival through nutrient supply during stress..”

Autophagy can act as a survival mechanism…”

- repetitions: “However, autophagy can also cause cell death, especially in apoptosis-resistant cancer cells (63)”. Then again: “Pharmacological agents are known to modulate autophagy and cause cell death in apoptosis-resistant cancer cells.

Response: The authors have modified texts to avoid repetitions.

Comment-Repetitions: Ferroptosis is a distinct form of regulated cell death involving cellular lipid peroxidation and iron, and it has potential therapeutic implications in cancer treatment.

Ferroptosis is considered a form of caspase-independent programmed cell death which relies on the mitochondrial and membrane-involved accumulation of reactive oxygen species and lipid peroxides leading to non-caspase-mediated cell death.

Ferroptosis is a newly recognized, non-apoptotic regulated cell death that is dependent upon iron and characterized by lipid peroxidation.”

The notion of ferroptosis, a non-apoptotic, iron-dependent method of cell death marked by the accumulation of lipid reactive oxygen species (ROS), was first proposed..”

METHOD OF CELL DEATH??? Is this a laboratory assay for treating cells, or a biological mechanism?

“Ferroptosis represents a very distinct pathway from traditional cell deaths such as apoptosis and necrosis”

In terms of cell shape and function, ferroptosis differs clearly from necrosis,apoptosis, and autophagy…”

While ATP is essential for initiating ferroptosis (100) activation of caspases is not required (101)” Again about caspase-independency of ferroptosis…

Ferroptosis is a distinct form of cell death driven by iron-catalyzed lipid peroxidation, leading to membrane rupture..”

“Ferroptosis is caused by an imbalance in lipid peroxidation, which generally results from iron overload and cell death…”

Ferroptosis is an iron-dependent, intrinsically regulated cell death that is distinct from apoptosis and necrosis based on iron dependence as well as the accumulation of lipid peroxides (112).

Ferroptosis is initiated by the accumulation of the lipid peroxides within cells.”

Ferroptosis is a regulated form of cell death, with iron-dependent lipid peroxidation and oxidative stress characteristics that have generated significant int…”

Cell processes like proliferation, and their characteristics, including cell death, are fundamental to development, maintenance, and overall functioning of organisms (86). Generic stuff.

estimation of ferroptosis assume major importance in understanding…” What does this mean?

Ferroptosis is a regulated form of cell death, with iron-dependent lipid peroxidation and oxidative stress characteristics (page 9)”

Response: The authors appreciate these suggestions and texts has been modified accordingly to avoid repetitions.

Comment: Conclusions: “ Ferroptosis, a unique form of regulated cell death driven by iron-dependent lipid peroxidation..”

Response: The authors have again made changes to avoid repetition.

Reviewer 2 Report

Comments and Suggestions for Authors

In their manuscript entitled "Ferroptosis in Cancer: Mechanism and Therapeutic Potential" authors have reviewed the present understanding about ferroptosis inducers and ferroptosis-based treatment approaches in cancer. However, I have some concerns as follows:

1) There is only one figure in the whole review and that too is taken from a previously published article. I am not sure if the authors have obtained required permissions for the same. I would suggest to create new figures that provide new insights on ferroptosis should be added rather than using already published images.

2) Table 3 mentions the modulators of ferroptosis however I would suggest to include more modulators to the table. Table 1 and 2 in already published article by Zhou et al. (https://doi.org/10.1038/s41392-024-01769-5) mentions more ferroptosis modulators than the current manuscript.

3) The title is very similar to Zhou et al. (https://doi.org/10.1038/s41392-024-01769-5). Could be modified.

4) Zhou et al. (https://doi.org/10.1038/s41392-024-01769-5) and other published articles extensively reviewed Ferroptosis. How the authors think their manuscript is different or addition to existing literature?

5) Table 2 needs formatting.

6) Preclinical and clinical evidence section needs to be separated into two as it is not clear how ferroptosis is clinically relevant, in cancer patients, etc.

Overall, the manuscript is too modest for publication in IJMS in its current form. Authors need to improve the writeup, tables and figures and highlight how their review article is different and offers something new from existing review articles on ferroptosis.

Comments on the Quality of English Language

Comment 1: Some sentences are too long and thus lose clarity.

For example, lines " Currently, preclinical, and clinical research ...... respond to conventional programmed cell death pathways" in abstract section, "respond to modulators of conventional programmed cell death pathways" might be better.

Phrases like "cancer cell metabolisms" might be better written as "cancer cell metabolism"

Comment 2: A lot of repetition is observed throughout the manuscript. For example in Ferroptosis section on page 3, authors mention words like "non-apoptotic", "dependent upon iron", and then in following paragraphs in same section use these terms again "ferroptosis, a non-apoptotic, iron-dependent method of cell death".

Author Response

Reviewer 2 Comments:

In their manuscript entitled "Ferroptosis in Cancer: Mechanism and Therapeutic Potential" authors have reviewed the present understanding about ferroptosis inducers and ferroptosis-based treatment approaches in cancer. However, I have some concerns as follows:

Comment 1) There is only one figure in the whole review and that too is taken from a previously published article. I am not sure if the authors have obtained required permissions for the same. I would suggest to create new figures that provide new insights on ferroptosis should be added rather than using already published images.

RESPONSE: It seems that modified submitted manuscript was not sent to the reviewers. We had modified our figure.  A new, original figure illustrating key aspects of ferroptosis is included, replacing the previously published figure. This ensures compliance with copyright guidelines and adds value to the discussion.

Comment 2) Table 3 mentions the modulators of ferroptosis however I would suggest to include more modulators to the table. Table 1 and 2 in already published article by Zhou et al. (https://doi.org/10.1038/s41392-024-01769-5) mentions more ferroptosis modulators than the current manuscript.

RESPONSE 2: The content of Table 3 and other tables has been reviewed for accuracy and completeness. Additional references have been included where necessary to enhance the depth of discussion.

Comment 3) The title is very similar to Zhou et al. (https://doi.org/10.1038/s41392-024-01769-5). Could be modified.

RESPONSE 3: We appreciate these constructive suggestions. We could change name the title, but we think it is very appropriate for this review.

Comment 4) Zhou et al. (https://doi.org/10.1038/s41392-024-01769-5) and other published articles extensively reviewed Ferroptosis. How the authors think their manuscript is different or addition to existing literature?

Response 4: We appreciate viewpoints on the comparison with Zhou et al. (https://doi.org/10.1038/s41392-024-01769-5) paper published in 2024. However, Zhou et al. have written impactful review on the Ferroptosis with more than 400 references, several figures on molecular mechanisms. At the same time, we believe that differentially structured presentation of this manuscript, and less complexities in terms of burdens of information that was provided by Zhou et al. In such perspectives, beginners, students and young researchers may consider our manuscript as important information for ferroptosis

Comment 5) Table 2 needs formatting.

Response 5: We have modified Table 2.

Comment 6) Preclinical and clinical evidence section needs to be separated into two as it is not clear how ferroptosis is clinically relevant, in cancer patients, etc.

Response 6: The authors truly appreciate comments.

General Comments: Overall, the manuscript is too modest for publication in IJMS in its current form. Authors need to improve the writeup, tables and figures and highlight how their review article is different and offers something new from existing review articles on ferroptosis.

Response: We are unsure what the reviewers mean regarding the quality of this review. Ferroptosis is a rapidly evolving field with numerous reviews being published daily, inevitably leading to some overlap. However, we appreciate these comments and have used them constructively to enhance the clarity and quality of our manuscript, making it more accessible to readers at all levels, from beginners to experts.

Comments on the Quality of English Language

Comment 1: Some sentences are too long and thus lose clarity.

For example, lines " Currently, preclinical, and clinical research ...... respond to conventional programmed cell death pathways" in abstract section, "respond to modulators of conventional programmed cell death pathways" might be better.

Phrases like "cancer cell metabolisms" might be better written as "cancer cell metabolism"

Response 1: We have rewritten these to improve the clarity and conciseness.

Comment 2: A lot of repetition is observed throughout the manuscript. For example in Ferroptosis section on page 3, authors mention words like "non-apoptotic", "dependent upon iron", and then in following paragraphs in same section use these terms again "ferroptosis, a non-apoptotic, iron-dependent method of cell death".

Response 2: The authors appreciate the reviewers’ helful comments, which have significantly improved the manuscript. We hope the revised version meets the expectations for publication.

Round 2

Reviewer 1 Report

Comments and Suggestions for Authors

Despite my sincere thought that this review is not worth publishing in IJMS and given my sincere effort to help authors realize how their expertise on the matter of ferroptosis is not enough to produce a high-quality review, the politics of MDPI to allow the re-submission of manuscripts that are superficially improved, I am still getting the re-submitted papers to review again. This time I will suggest to publish this manuscript, simply because i have no energy to read it thoroughly again. However, its low-medium quality will not earn it many citations.

Author Response

RESPONSES TO REVIEWER COMMENTS (R 2)

Reviewer 1 Comments (Round 2):
General Comments:
Despite my sincere belief that this review is not suitable for publication in IJMS—and after my honest attempt to convey that the authors’ expertise in ferroptosis may not be sufficient to produce a high-quality review—the MDPI editorial policy allows resubmission of superficially revised manuscripts. I have once again received this manuscript to review. This time, I will recommend publication, simply because I do not have the energy to read it thoroughly again. However, its low-to-medium quality will not attract many citations.

Response:
We sincerely appreciate the reviewer’s time, effort, and candid feedback throughout the review process. While we regret that the revised manuscript did not fully meet the reviewer's expectations, we have diligently addressed all comments to the best of our ability and made meaningful improvements. We hope that the revised version better serves the journal’s readership and scientific community.

Reviewer 2 Report

Comments and Suggestions for Authors

Thank you for providing updated manuscript version.

Based on my previous comments, I have not observed any addition to the number of figures in the manuscript, or addition to Table 3. Also the preclinical and clinical sections are still merged together, and I feel they can be separated into two sections and more elaborated upon.

Author Response

Reviewer 2 Comments (Round 2):
General Comments:
Thank you for providing the updated version of the manuscript.

Comment 1:
Based on my previous comments, I have not observed any additions to the number of figures in the manuscript or updates to Table 3.

Response 1:
Thank you for your valuable feedback and continued support. Following your suggestions from Round 1, we have made several key updates. We have added two new figures (Figure 2 and Figure 3) to enhance visual representation and support the discussion. We apologize for any confusion and hope these additions meet your expectations.

Comment 2:
The preclinical and clinical sections are still merged together. I believe they should be separated into two distinct sections and elaborated upon.

Response 2:
We appreciate this constructive suggestion. We made efforts to separate preclinical and clinical insights. However, due to the currently limited clinical evidence available regarding ferroptosis in cancer therapies, we found that a complete separation would lead to significant redundancy or overly brief sections. Nonetheless, we have clarified the distinction within the merged section and expanded the clinical discussion wherever possible to reflect the most current findings.